

# Quantifying the effect of seasonal and vertical habitat tracking on planktonic foraminifera proxies

Lukas Jonkers* and Michal Kučera

MARUM | Universität Bremen, Leobenerstraße 8, Bremen, Germany.

* correspondence to ljonkers@marum.de

Keywords: planktonic foraminifera, seasonality, depth habitat, proxies

Key points:

- PF depth and seasonal habitat change with temperature

- PF proxy signals therefore underestimate gradients in space and time

- Depth and seasonal habitat variability can be predicted and thus accounted for





## Abstract

The composition of planktonic foraminiferal (PF) calcite is routinely used to reconstruct climate

change and variability. However, PF ecology leaves a large imprint on the proxy signal. The seasonal

and vertical habitat of planktonic foraminifera (PF) species varies spatially, causing variable offsets

from annual mean surface conditions recorded by sedimentary assemblages. PF seasonality changes

with temperature in a way that minimises the environmental change that individual species

experience. While such habitat tracking could lead to an underestimation of spatial or temporal

trends and variability in proxy records, most paleoceanographic studies are based on the assumption

of a constant habitat. Although the controls on depth habitat variability are less well constrained, it is

not unlikely that habitat tracking also affects PF depth habitat. Despite the implications, the effect of

this behaviour on foraminifera proxy records has not yet been formally quantified on a global scale.

Here we attempt to characterise the effect of habitat tracking on the amplitude of environmental

change recorded in sedimentary PF using core top $\delta^{18}O$ data from six species, which we compare to

predicted $\delta^{18}O$. We find that the offset from mean annual near-surface $\delta^{18}O$ values varies with

temperature, with PF $\delta^{18}O$ indicating warmer than mean conditions in colder waters (on average by -

0.1‰ (or 0.4°C) per °C), thus providing a first-order quantification of the degree of underestimation

due to habitat tracking. We then use an empirical model to estimate the contribution of seasonality

to the observed difference between PF and annual mean $\delta^{18}O$ and use the residual $\Delta\delta^{18}O$ to assess

trends in calcification depth. Our analysis indicates that in all species calcification depth increases

with temperature. Consistent with hydrographic conditions, vertical habitat adjustment is dominant

in tropical species, whereas cold-water species mainly changes their seasonality when tracking their

'optimum' habitat. Assumptions of constant PF depth or seasonal habitat made when interpreting

proxy records are thus invalid. The approach outlined here can be used to account for these effects,

enabling more accurate reconstructions and improved data-model comparison.





# 1. Introduction

The chemical composition of planktonic foraminifera shells reflects the environmental conditions in which they precipitate. Fossil shells therefore serve as the prime source of information about the past

state of the oceans. Because the seasonal flux and depth habitat of planktonic foraminifera species are not constant in space and time, accurate reconstructions require an understanding of how species ecology influences the climate signal preserved in the fossil record. Planktonic foraminifera inhabit a wide vertical range of the water column and often show distinct variability in their seasonal abundance (e.g. Field, 2004; Tolderlund and Bé, 1971; Fairbanks et al., 1980; Jonkers et al., 2010;

Jonkers et al., 2013; Deuser et al., 1981). Therefore, rather than reflecting annual mean surface conditions the average proxy signal in sedimentary planktonic foraminifera is weighted towards conditions at the depth and season of calcification (Mix, 1987). While species-specific seasonality and calcification depth are often taken into account, it is implicitly assumed that both remain constant in time and space. This assumption is at odds with observations from the present-day ocean and this

could have important implications for reconstructions of climate change and inferences of climate sensitivity.

Plankton tow and sediment trap studies have shown large variability in the phenology within individual planktonic foraminifera species (e.g. Tolderlund and Bé, 1971; Zaric et al., 2005). A recent review demonstrated that this variability is widespread and follows a predictable pattern consistent

with the concept that foraminifera track their optimum habitat (Jonkers and Kučera, 2015). Two broad ecological groups with different seasonality patterns were found; it was shown that outside the tropics, warm-water species narrow their occurrence into the season of maximum temperatures. The seasonality in cold-water species also shows a clear relationship with temperature as their flux peak generally occurs earlier in the year in warmer waters (Jonkers and Kučera, 2015). While the latter

trend appears to be driven by the timing of food availability rather than reflecting the thermal tolerance of the species, both patterns have the same effect on the fossil record as they reduce the amplitude of (temperature) change that bulk samples of their fossil shells will reflect. This indicates that changes in seasonality need to be taken into account when interpreting proxy records.



The depth habitat of planktonic foraminifera species also shows clear variability in space and time

(Field, 2004; Fairbanks and Wiebe, 1980; Schiebel et al., 2001; Peeters and Brummer, 2002; Rebotim

et al., 2016). Food and light availability, (thermal) stratification and temperature have all been

suggested to be potential drivers of the vertical distribution of planktonic foraminifera (Schiebel et al.,

2001; Fairbanks and Wiebe, 1980; Ortiz et al., 1995; Salmon et al., 2015; Fairbanks et al., 1982). In

contrast to seasonality a global overview is lacking and the exact controls on depth habitat variability

within species remain poorly constrained. The issue of changing vertical habitat is further complicated

by the tendency that many foraminifera likely migrate in the water column during their life and add

proportionally more calcite at later stages in their life, potentially resulting in a mismatch between

depth habitat and calcification depth (e.g. Duplessy et al., 1981). Whereas depth habitat can be

directly observed, calcification depth is generally estimated from geochemical data and hence more

uncertain. Nevertheless, depth habitat and calcification depth are related, as deeper dwelling species

will also have a greater calcification depth. Here we hypothesise that, similar to seasonality, the depth

habitat and therefore calcification depth is related to temperature and that changes in temperature

will lead to adjustments in depth habitat such that the environmental changes planktonic

foraminifera experiences during their life cycle are minimised.

The combined effect of seasonal and depth habitat tracking would be that temporal and spatial

gradients in planktonic foraminiferal proxy records are reduced compared to the gradients in the

mean annual value of the reconstructed parameter. For instance, in the case of temperature, it would

lead to positive offsets from annual mean near-surface temperatures at times of cooling as planktonic

foraminifera shift their seasonal and depth habitats to the warmer season and/or to shallower

depths. The partitioning of this effect into seasonality and depth habitat likely varies by region,

depending on the ratio of seasonal over vertical temperature variability in the upper water column

(Fig. 1). This implies that for tropical species constraining the depth habitat will be more important

than seasonality, whereas the opposite is true for species living in mid- and high latitudes.

Variability in seasonal and vertical habitat within individual species is well known and the dependency

of foraminifera habitat on climate has been alluded to before (Ganssen and Kroon, 2000; Mix, 1987;

Mulitza et al., 1998; Jonkers and Kučera, 2015). In addition, several modelling studies have

investigated the potential dampening effect of seasonality (Fraile et al., 2009a; Fraile et al., 2009b;



Kretschmer et al., 2016). Yet a systematic quantification of the effect of habitat tracking on planktonic

foraminifera proxies based on observational evidence, as well as an assessment of the respective

roles of seasonality and depth habitat, is lacking. Here we use core top stable isotope data to first

demonstrate that foraminifera proxies are indeed affected by habitat tracking. We then show that

this effect can be parametrised and assess the relative importance of variable seasonality and depth

habitat. Our findings have important implications for the interpretation of paleoceanographic records

and could help to bridge the gap between paleoceanographic data and model simulations.

## 2. Data and approach


If shifts in depth and seasonal habitat would act to minimise the change in the ambient environment

of the planktonic foraminifera, then the proxy signal preserved in their shells should show an offset

from mean annual values that varies with temperature. To test this conjecture we compare core top

stable oxygen isotope ($\delta^{18}O$) values from different species with seasonally and vertically resolved

estimates of equilibrium $\delta^{18}O$. We use quality controlled data with strict age control (chronozone 1-4)

from the MARGO core top dataset (Waelbroeck et al., 2005). This dataset contains data for six species

of planktonic foraminifera: *Trilobatus sacculifer* (n = 38), *Globigerinoides ruber* (white (n = 131) and

pink (n = 20) varieties), *Globigerina bulloides* (n = 131), *Neogloboquadrina incompta* (n = 46) and

*Neogloboquadrina pachyderma* (n = 253). We exclude samples from the Mediterranean from our

analysis because of difficulties of estimating seawater $\delta^{18}O$ in this evaporative basin, and removed *T.*

*sacculifer* data from Pacific Ocean sites deeper than 3 km as these are thought to be affected by

dissolution (Wu and Berger, 1989).

We compare the planktonic foraminifera $\delta^{18}O$ ($\delta^{18}O_{foram}$) to predicted $\delta^{18}O$ ($\delta^{18}O_{eq}$) calculated using

$\delta^{18}O$-temperature equation by Kim and O'Neil (1997). Following the approach of LeGrande and

Schmidt (2006) we estimate seawater $\delta^{18}O$ using regionally defined salinity-$\delta^{18}O_{sw}$ relationships for

the upper 200 m using the Global Seawater Oxygen-18 Database (Schmidt et al., 1999). Conversion

from the SMOW to PDB scale was done by subtracting 0.27 ‰ (Hut, 1987). Temperature and salinity

data were taken from the World Ocean Atlas 2001 (Boyer et al., 2002; Stephens et al., 2002) and area

weighted averages were obtained from the four 1 degree areas surrounding each core top position.





We start with comparing the observed $\delta^{18}O_{foram}$ to annual mean $\delta^{18}O_{eq}$ for the upper 50 m as this is

the depth interval where these species are most likely to calcify.

## 3. Habitat tracking in planktonic foraminifera

The observed $\delta^{18}O$ of all species show deviations from expected mean annual $\delta^{18}O_{eq}$ by up to 3 ‰

(Fig. 2). If our hypothesis of habitat tracking holds, the $\Delta\delta^{18}O$ ($\delta^{18}O_{foram}$ - $\delta^{18}O_{eq}$) should show a

positive relationship with temperature. Indeed, the $\Delta\delta^{18}O_{annual.mean}$ values of the three tropical species

(*G. ruber* (white and pink) and *T. sacculifer*) show a positive relationship with mean temperature (Fig.

2). The slopes vary between 0.04 and 0.14 ‰ °C$^{-1}$. In general, $\Delta\delta^{18}O_{annual.mean}$ values are close to 0 at

high temperatures and negative $\Delta\delta^{18}O_{annual.mean}$ values, indicating higher calcification temperatures,

occur in colder waters.  Among the cold-water species, *N. incompta* displays a consistent positive

relationship between temperatures and $\Delta\delta^{18}O_{annual.mean}$ above ~5 °C, whereas below this temperature

the relationship has the opposite sign (Fig. 2). These observations at low temperature stem from

samples in the Nordic Seas outside of the direct path of the North Atlantic Drift and we suspect that

these (partly) reflect right-coiling variants of *N. pachyderma* (Bauch et al., 2003) and we have

therefore excluded them from further analysis. $\Delta\delta^{18}O_{annual.mean}$ values of *N. pachyderma* are generally

positive and show an increased spread towards higher values in warmer waters (Fig. 2). *G. bulloides* is

the only species that does not show any trend in $\Delta\delta^{18}O_{annual.mean}$; modal values are close to 0, but the

distribution is skewed towards positive offsets (Fig. 2). This species was therefore excluded from

further analyses. In summary, five of the six analysed species appear to minimise experienced

temperature/environmental change, consistent with our hypothesis that habitat tracking affects

planktonic foraminifera proxies.

## 4. Seasonality

Next, using simple empirical models for seasonality we assess how much of the trend in

$\Delta\delta^{18}O_{annual.mean}$ could be due to changes in seasonality alone. To this end we calculate a flux-weighted

$\delta^{18}O_{eq}$ ($\delta^{18}O_{season}$) for the upper 50 m of the water column using a simple seasonality model. Based on



previous work (Jonkers and Kučera, 2015) we describe the $\log_{10}$-transformed flux pattern as a sine

wave of which we change the amplitude and phasing as a function of mean annual temperature. For

tropical species we fix the peak in the flux in September (March in the southern hemisphere), which is

generally the warmest month, and increase the amplitude linearly with decreasing temperature with

a species-specific slope derived from sediment trap data (Fig. 3; Table 1). While this model does not

account for the random peak flux timing at high temperatures (Jonkers and Kučera, 2015), the

seasonal amplitude of the shell flux and of $\delta^{18}O_{eq}$ are very small at these temperatures, and the model

serves as a reasonable approximation of the seasonality pattern that characterises this species group.

For cold-water species we fix the amplitude at the average value for this group (0.66) and vary the

timing of the peak flux as a function of temperature (Fig. 3). Below a critical low temperature we set

the peak timing to September and above a critical high temperature to March (reversed for Southern

hemisphere); between these temperatures, the modelled flux pattern has two peaks a year that

linearly shift towards earlier in the year in colder waters (table 2). While simple, this model represents

a realistic scenario, derived from observations and can thus be applied to all of the studied species.

We gauge the effect of the flux weighting by determining i) the (change in the) spread of the $\Delta\delta^{18}O$

values and ii) the slope of the $\Delta\delta^{18}O$-temperature relationship.

Accounting for seasonality using this model reduces the root means square error (RMSE) in the $\Delta\delta^{18}O$

values of *G. ruber* (pink) by 21% and the slope by 37% (Fig. 4). For *G. ruber* (white) the values are 12

and 77 % respectively (Fig. 4). Due to large positive $\Delta\delta^{18}O$ values for *T. sacculifer* at high

temperatures, flux-weighting has a negligible effect on the spread in the values (1 %), but it reduces

the slope by 22 % (Fig. 4). The values for *N. incompta* are 49 and 59 % and for *N. pachyderma* 17 and

36 % (Fig. 4). For none of the species this adjustment for seasonality leads to an increase in the spread

of the $\Delta\delta^{18}O$ values, on the contrary, in most cases the predicted $\delta^{18}O_{eq}$ are closer to the $\delta^{18}O_{foram}$.

This indicates that even by using a simple empirical model for seasonality, predictions of the fossil

signal can be improved, as long as the mean annual temperature is constrained from independent

data.





## 5. Calcification depth

In none of the species investigated here, the adjustment for seasonality completely removes the relationship between $\Delta\delta^{18}O$ and temperature. Therefore, one may assume that at least a part of the relationship could reflect an adjustment of calcification depth. To investigate if the trends in the

$\Delta\delta^{18}O_{season}$ reflect an increase of calcification depth towards the tropics, we determine the depth at which $\Delta\delta^{18}O_{season}$ is smallest and assess if there is a relationship between this apparent calcification depth and mean annual temperature. This analysis reveals that of the tropical species *G. ruber* (pink) shows the shallowest (apparent) calcification depth, followed by *G. ruber* (white) and *T. sacculifer* (Fig. 5). All species show an increase in calcification depth with temperature (even though the scatter

is large). This rules out that the relationships between $\Delta\delta^{18}O$ and temperature (Fig. 2 and 4) reflect calcification at a constant, but greater depth than in the near surface layer. Rather, this correlation is consistent with the hypothesis that planktonic foraminifera (passively) track an optimum vertical habitat. *N. incompta* has variable calcification depths that show a steep slope with temperature (Fig. 5). The positive $\Delta\delta^{18}O_{season}$ values of *N. pachyderma* indicate a calcification depth consistently below

50 m (Fig. 5).

Next we use the linear relationships between apparent calcification depth and temperature (Fig. 5) to explain the fossil signal. We thus adjust the $\delta^{18}O_{season}$ to a depth-specific signal, using the depth-temperature relationship identified earlier (Fig. 5) to calculate $\Delta\delta^{18}O_{season.depth}$. In *G. ruber* (pink) this leads to a further 50 % reduction in the RMSE and a $\Delta\delta^{18}O_{season.depth}$-temperature slope that is close to

0 (Fig. 6). In *G. ruber* (white) the reduction in the spread in the data is more modest (12 %) and the slope is reduced to 0 (Fig. 6). For *T. sacculifer* also only modest additional reductions are achieved: 8 and 11 % for RMSE and slope, respectively (Fig. 6). Among the cold-water species *N. incompta* shows the clearest relationship between $\Delta\delta^{18}O_{season.depth}$ and temperature and adjustment for calcification depth yields a reduction of the RMSE of 8 and of the slope of 46 % (Fig. 6). In *N. pachyderma* no

further reduction in the slope is achieved and the RMSE decreases by 22 %.



## 6. Seasonality vs. depth habitat

Our analysis allows partitioning of habitat change into changes in seasonality and calcification depth for species where temperature seems important for determining their habitat. In general, the improvement of the prediction of the $\delta^{18}O$ is larger for the slope of the $\Delta\delta^{18}O$-temperature

relationship than for the spread in the $\Delta\delta^{18}O$ values (Fig. 7). This may point to some degree of inherent noise in the observations (e.g. related to different size fractions used for the measurements (Friedrich et al., 2012)), or it could also be due to uncertainty in the $\delta^{18}O_{eq}$ values, which are based on climatology and salinity-based estimates of $\delta^{18}O_{sw}$. Moreover, the noise may also reflect the simplicity of the seasonality model we have used. Nevertheless, *G. ruber* (pink) and *N. incompta* show coherent

behaviour with respect to both parameters (Fig. 7). For *N. incompta* seasonality explains most of the trend in $\Delta\delta^{18}O_{annual.mean}$, whereas for *G. ruber* (pink) depth habitat appears more important. This is consistent with their distribution: *N. incompta* predominantly inhabits high and mid latitudes where seasonal temperature change is larger than vertical temperature gradients and *G. ruber* (pink) is restricted to the tropics where the opposite situation prevails (Fig. 1). This pattern provides support

for our approach and suggests that both seasonality and depth habitat variability are important for interpretation of the proxy signal preserved in the sediment. The picture is less clear for *G. ruber* (white) and *T. sacculifer*. For the latter species the improvement in the prediction of their $\delta^{18}O$ is generally smaller, which may be due to a remnant dissolution signal at the high temperature end of the species distribution in the Pacific. For *G. ruber* (white), the signal/noise ratio in the data appears

lower than in the other species, which may reflect a disproportionate effect of secondary variables, such as changing proportionality and inconsistent recognition of the ecologically distinct morphotypes (Steinke et al., 2005) that are now assigned to different taxa (Aurahs et al., 2011).

An important caveat in the attribution of the improvement in the prediction of the fossil proxy signal

to either seasonality or calcification depth is the simplicity of the seasonality model used. While flux patterns of planktonic foraminifera can be reasonably approximated by a sine function (Jonkers and Kučera, 2015), it is important to realise that this is only an approximation and seasonal flux pulses are often narrower and more focussed, leading to flux-weighting to a shorter period within the year. The



model used here is therefore a conservative estimate of the importance of seasonality. Next, implicit

in our approach is the assumption that planktonic foraminifera form their skeleton accordance with

inorganic calcite precipitation and that their $\delta^{18}O$ can be described using the equation by Kim and

O'Neil (1997). While this appears to be the case for some species (Jonkers et al., 2010; Jonkers et al.,

2013; Loncaric et al., 2006), there are also indications that, in particular for tropical species, different

equations are more appropriate (Mulitza et al., 2003; Spero et al., 2003). Species-specific

paleotemperature equations proposed by the latter authors have a non-quadratic form, but almost

identical slopes as the Kim and O'Neil (1997) equation yet are offset by 0.3-0.6 ‰, with the offset

increasing with temperature. For instance, using the Mulitza et al. (2003) equation for *T. sacculifer*

would lead to more positive $\Delta\delta^{18}O_{annual.mean}$ values and slightly steeper $\Delta\delta^{18}O$-temperature

relationships (Fig. 8). This suggests a generally greater calcification depth and would change the

attribution of depth habitat and seasonality influence, rendering depth habitat more important (Fig.

7). However, it would not affect our main conclusion that the proxy signal of planktonic foraminifera

is affected by habitat tracking.

## 7. Discussion

Five out of six species analysed show a temperature dependency of the offset between $\delta^{18}O$ of the

foraminiferal shells and the annual mean $\delta^{18}O$ of the upper water column (Fig. 2). In addition, these

species show a positive relation between apparent calcification depth and temperature (Fig. 5).

Together, these observations provide a strong indication that temperature, either directly or by acting

on other temperature-related variables, causes changes in the habitat of foraminifera. Such an

important role for temperature in determining the vertical and seasonal habitat is not unexpected

given that temperature appears to be dominant in controlling the spatial distribution of species

(Morey et al., 2005; Bé and Hutson, 1977), their flux (Zaric et al., 2005) and seasonality (Jonkers and

Kučera, 2015) and appears important for test growth (Lombard et al., 2009).

Several studies have shown that formation of secondary calcite layers (e.g. gametogenic calcite or a

crust) at the end of the life of a specimen, presumably deep in the water column could be responsible

for higher $\delta^{18}O$ of sedimentary foraminifera compared to those collected in the upper water column



(Duplessy et al., 1981; Bé, 1980). To the best of our knowledge there is no evidence that such

secondary calcite is formed with a different isotopic (dis)equilibrium than the lamellar calcite. We

therefore assume that our inferences are not affected by differences in calcification during ontogeny.

Nevertheless, the addition of such a crust in deeper (colder) waters could in principle lead to the

observed increase in apparent calcification depth with temperature because of steeper vertical

temperature gradients in the tropics. However, foraminifera grow their tests exponentially and the

last chambers that make up most of the test mass are formed in the last few days of their life,

presumably close to the time of the secondary calcite formation (Bé, 1980). The compositional

contrast between the bulk of the lamellar calcite and the crust calcite is thus likely to be smaller than

estimated from the comparison of surface tows and sediment (cf. Jonkers et al., 2016). Consequently,

the apparent calcification depth we infer here likely incorporates this effect and the increase in

apparent calcification depth that we observe most likely reflects homeostatic habitat adjustment.

Next to temperature and $\delta^{18}O_{seawater}$ the $\delta^{18}O$ of foraminiferal calcite is to a lesser degree also

influenced by the $CO_3^{2-}$ concentration in seawater (Spero et al., 1997). Because of the generally

positive correlation between temperature and $[CO_3^{2-}]$ in seawater, the trends we observe in

$\Delta\delta^{18}O_{annual.mean}$ (Fig. 2) could be dampened by a $CO_3^{2-}$ influence. However, the $CO_3^{2-}$ effect is only

modest (0.002 ‰ $\mu mol^{-1}$ $kg^{-1}$) and to fully account for the on average 1‰ difference we observe over

the temperature range in our dataset, unrealistically large gradients in $[CO_3^{2-}]$ would be required. The

trends thus most likely dominantly reflect real changes in the habitat of planktonic foraminifera.

While the majority of the species investigated here show clear indications of temperature-dependent

depth and seasonal habitat variability, the picture for *N. pachyderma* is less clear. In the species most

of the trend in $\Delta\delta^{18}O_{annual.mean}$ values appears driven by an increased spread in $\Delta\delta^{18}O$ at higher

temperatures (Fig. 2). Some of these values are unrealistically large and stem from observations in the

northern North Atlantic south of 50° N, thus outside the general distribution range of the species. This

suggests that these observations may reflect expatriated specimens that calcified in colder regions or

may point to inaccuracies in the chronological control and reflect (partly) shells of glacial age.

Alternatively, these samples could be affected by admixture of sinistrally coiled *N. incompta* (Darling



et al., 2006). It is puzzling though that the effect of seasonality is not larger since the species shows a

clear latitudinal shift in the timing of the peak flux (Jonkers et al., 2010; Jonkers et al., 2013; Jensen,

1998; Wolfteich, 1994; Kohfeld et al., 1996). However, the species is also known to inhabit a broad,

but generally deeper, zone of the upper water column (Carstens et al., 1997; Pados and Spielhagen,

2014) where seasonal temperature is smaller than in the near surface layer, possibly rendering a

seasonality effect difficult to detect.

At face value, the absence of a $\Delta\delta^{18}O_{annual.mean}$-temperature trend in *G. bulloides* may suggest that this

species holds the best promise of providing reconstructions of mean annual near surface conditions

(Fig. 2). However, the distribution of $\Delta\delta^{18}O_{annual.mean}$ is noisy, suggesting that caution is required to

interpret the species proxy signal. Similar to *N. pachyderma* this species also shows clear latitudinal

changes in seasonality (Jonkers and Kučera, 2015; Tolderlund and Bé, 1971). However, *G. bulloides* is

characterised by considerable cryptic diversity (Darling and Wade, 2008). Possible genotypic

ecological differences could therefore obscure ecological patterns at the morphospecies level.

Alternatively, being an opportunistic species, depth and seasonal habitat variability of *G. bulloides*

may be driven by other parameters than temperature. Indeed, previous studies have shown that the

distribution of this species is driven by food availability (Schiebel et al., 1997; Jonkers and Kučera,

2015). Whether or not the species shows habitat tracking and how this would affect its fossil record

remains unclear, but we caution that the result of our study cannot be taken to indicate that proxy

records from this species record the actual magnitude of environmental change.

Since planktonic foraminifera seasonality and calcification depth appear to be affected by habitat

tracking, our ability to accurately reconstruct past ocean properties would benefit from improved

understanding of the drivers of their habitat variability. In particular, the controls on depth (and

calcification) habitat remain poorly constrained. Due to logistical challenges, very few studies exist

that have attempted to systematically investigate depth habitat variability. On the other hand, the

realisation of the importance of habitat homeostasis in planktonic foraminifera could help to

formulate more realistic, mechanistic models of planktonic foraminiferal distribution in time and

space (e.g. Lombard et al., 2011) and further improve our capabilities of forward proxy modelling. At

any rate, the observations and the simple conceptual modelling exercise shown here serve as



reminder that assumptions of constant seasonality and depth habitat underlying many

paleoceanographic studies are not valid and the implications thereof are likely to be substantial. Our

analysis indicates that an observed change in a proxy value reflects a change in the climate state as

well as a change in the species habitat.

## 8. Implications

Habitat tracking behaviour of planktonic foraminifera has important implications for

paleoceanographic reconstructions. It may suggest that the temperature niche of planktonic

foraminifera inferred from their abundance in the sediment (e.g. Kucera, 2007) may be overestimated

since their occurrence is not driven by mean annual sea surface temperature, but rather by whether

their temperature niche is realised at any depth or season. It should thus be possible to define

planktonic foraminifera temperature ranges (sensitivity) more precisely, which may help to improve

transfer functions and is important for understanding of their ecology.

Another consequence of habitat tracking is that spatial and temporal differences reflected in the

sedimentary foraminifera represent an underestimation of the actual gradients in the mean

conditions, because temperature change forces the foraminifera to live in a seasonal or vertical

'window' where conditions are closest to optimal (cf. Jonkers and Kučera, 2015). We observe

considerable variability in the slope of the $\Delta\delta^{18}O_{annual.mean}$-temperature relationships, but the average

for the four species that show the clearest signal (*G. ruber* (pink and white), *T. sacculifer* and *N.

incompta*) is 0.1 ‰ °C$^{-1}$ (Fig. 2). This is equivalent to a 40 % (0.4 °C °C$^{-1}$) underestimation of

reconstructed temperature change.

The existence of such underestimation can be observed through comparison of time series of

different temperature proxies. Previous studies have shown that Holocene temperature trends and

temperature variability inferred from foraminiferal Mg/Ca ratios are generally of lower magnitude

than those derived from alkenone unsaturation indices (Gill et al., 2016; Leduc et al., 2010). While it is

not a priori clear that the alkenone signal is unaffected by seasonal habitat variability of

coccolithophores (Rosell-Melé and Prahl, 2013), this comparatively low variability inferred from

planktonic foraminifera proxies provides support that habitat tracking minimises amplitude of the



recorded environmental change. Comparison of Mg/Ca-derived and transfer function based

temperature evolution across the deglaciation provides further indications that habitat tracking

dampens the foraminifera proxy signal (Fig. 9). While both proxies indicate a clear warming step

during the deglaciation, the amplitude of the Mg/Ca-based estimate is significantly lower. In addition,

the single species Mg/Ca-temperature estimate lacks the smaller cooling and warming trends seen in

the transfer function-based estimate during the glacial and Holocene respectively. Using the linear

$\Delta\delta^{18}O_{annual.mean}$-temperature relationships (Fig. 2) we also predicted the *G. ruber* (pink) temperature

signal assuming that the assemblage-based temperatures represent an accurate estimate of mean

annual temperature and using a conversion from $\delta^{18}O$-temperature sensitivity of 0.25 ‰ °C$^{-1}$ (Fig. 9).

The high degree of agreement between the predicted and observed temperature evolution provides

quantitative support for the idea that habitat tracking reduces the amplitude of the foraminifera

proxy signal.

Accounting for the dampening effect due to habitat tracking would likely increas the magnitude of

reconstructed climate change as well as estimates of climate variability on longer time scales. This

could have profound implications for inferred climate dynamics; it may mean, for instance, that

estimates of climate sensitivity (e.g. Snyder, 2016) may be too low (or at least that the uncertainty of

the estimate can be reduced). In addition, model-data comparison indicates that climate models

systematically underestimate temperature variability (Laepple and Huybers, 2014), which has

implications for both attribution of ongoing climate change as well as for climate predictions. Since

habitat tracking dampens variability in the foraminifera proxy record, the mismatch between

modelled and reconstructed climate variability may be even larger.

It is important to note that habitat tracking would not only affect stable isotope records and Mg/Ca-

based temperature estimates, but any geochemical proxy based on planktonic foraminifera. However,

the size of the effect will depend on the magnitude of the seasonal and vertical gradients in the

parameters that are inferred. Deconvolving the effect of habitat tracking into seasonality and

calcification depth effects in data from the fossil record is however not straightforward.  For instance,

minor changes in mean temperature may be accommodated by changes in the habitat of foraminifera

and remain invisible. Properly accounting for habitat tracking in paleoceanographic records can thus

only be achieved using a multiproxy and iterative approach. However, the effect can be taken into



account in model-data comparison efforts through improved prediction of seasonality and depth

habitat using modelled hydrography. This can be achieved by applying the simple empirical

relationships identified here, or by using more complex mechanistic models of foraminifera

distribution (Lombard et al., 2011; Fraile et al., 2008).

## 9. Conclusions

Through comparison of observed and predicted $\delta^{18}O$ data of six common planktonic foraminifera we

have demonstrated that the average geochemical signal preserved in a population of fossil shells

shows a temperature-dependent offset from mean annual sea surface conditions. This most likely

reflects shifts in the seasonal and depth habitat in response to temperature, or temperature-related

environmental, changes (Fig. 9). As a consequence of this homeostatic behaviour, the fossil record of

these species, and likely also of others, does not reflect the full range of climate variability. Our

analysis indicates that spatial and temporal gradients in temperature may be underestimated by 40

%, clearly highlighting the need to account for climate-dependent habitat variability in the

interpretation of paleoceanographic records based on planktonic foraminifera. Using a simple

empirical model we attempted to assess the relative influence of seasonality and depth habitat

variability. While improvements to this empirical approach are possible, we observe species-specific

partitioning of depth habitat versus seasonality that appears consistent with oceanographic

conditions within their areal distribution. In the tropical species *G. ruber* (pink) we find that habitat

tracking is primarily due to adjustments in the calcification depth. This is in agreement with the larger

vertical than seasonal temperature gradients in the tropics. The offsets from annual mean surface

conditions in *N. incompta*, on the other hand, appear dominantly driven by changes in the

seasonality, consistent with the dominance of seasonal over vertical temperature variability in the

regions where it occurs. Our data underscore the importance of ecology in setting the climate signal

preserved in fossil foraminifera. The recognition of predictable habitat tracking will help to improve

the accuracy of paleoceanographic reconstructions and aid model-data comparison.





## 395 Acknowledgments

We thank Stefan Mulitza and Thomas Laepple for valuable discussions, which helped to improve this manuscript. LJ was supported by the German climate modelling initiative PalMod funded by the Federal Ministry of Education and Research (BMBF). The data used in this study are all in the public domain, but a data sheet is provided as electronic supplement. R (R core team, 2016) code is available

upon request from LJ.

## Tables

Table 1: temperature-amplitude relationships for the modelled flux pattern of tropical species based on Jonkers and Kucera (2015).

| Species | intercept | slope |
|---------|-----------|-------|
| *G. ruber* (pink) | 2.16 | -0.07 |
| *G. ruber* (white) | 0.99 | -0.02 |
| *T. sacculifer* | 0.85 | -0.02 |

Table 2: critical temperatures (°C) that determine the phasing of the shell flux of cold-water species. Between these two temperatures the flux pattern is characterised by two peaks a year that shift as a function of temperature to earlier in the year in warmer water (Jonkers and Kučera, 2015).

| Species | T.crit.lo | T.crit.hi |
|---------|-----------|-----------|
| *N. incompta* | 9 | 15 |
| *N. pachyderma* | -5 | 7 |

## Figure captions

Fig. 1: Distribution of core top $\delta^{18}O$ data used in this study. Background colours represent the $\log_{10}$-ratio of the temperature range at the surface to the temperature range in the annual mean values between 0 and 200 m depth. Blue colours thus indicate areas where seasonal temperature gradients





are larger than vertical gradients and red colours indicate the opposite. The thin black contour line

shows the zero level of this ratio.


Fig. 2: Offset between predicted annual mean near surface and observed $\delta^{18}O$. All species except *G.*

*bulloides* show a trend in $\Delta\delta^{18}O_{annual.mean}$ values with mean annual temperature in the upper 50 m

($MAT_{0-50m}$) of the water column suggesting that planktonic foraminifera adjust their habitat to

minimise temperature change in their environment. Histograms show the spread in the

$\Delta\delta^{18}O_{annual.mean}$ values. The root mean square error (RMSE) and the linear slope (m) of the $\Delta\delta^{18}O$-

temperature relationship are indicated in the upper left corner of each panel. The grey dots in the

panel for *N. incompta* show the data that are excluded from further analyses as they most likely stem

from right-coiling morphotypes of *N. pachyderma*.

Fig. 3: Schematic representation of the seasonality model. Upper panels show the annual flux

patterns; colours indicate temperature, where blue is cold and red is warm. Lower panels show the

timing of the peak in the year. For a more detailed explanation of the model see §4 and tables 1 and

2.

Fig. 4: Offset between flux-weighted predicted and observed $\delta^{18}O$. Grey symbols represent

$\Delta\delta^{18}O_{annual.mean}$. Note the general reduction in the spread of the data (RMSE) and slope of the $\Delta\delta^{18}O$-

temperature relationship (m) compared to $\Delta\delta^{18}O_{annual.mean}$ (Fig. 2).

Fig. 5: Relationship between apparent calcification depth (ACD) and temperature. Data are

summarised in 2-degree bins and error bars represent standard deviations within each bin. The data

points at the cold temperature end of *G. ruber* (white) are excluded since these are more likely to

reflect outliers or advected specimens.

Fig. 6: Offset between flux-weighted and depth-adjusted predicted and observed $\delta^{18}O$. Grey symbols

represent $\Delta\delta^{18}O_{annual.mean}$.

Fig. 7: Partitioning of the improvement in the prediction of the fossil $\delta^{18}O$ signal into seasonality and depth habitat for both RMSE of $\Delta\delta^{18}O$ and the slope between $\Delta\delta^{18}O$ and $MAT_{0\text{-}50m}$. Colours denote species and the size of each dot is proportional to the total improvement achieved. The open circles

illustrate the partitioning for *T. sacculifer* using the paleotemperature equation of Mulitza et al. (2003).

Fig. 8: Assessing the effect of the use of a different paleotemperature equation. The panels show the same as Fig. 2, 4, 5 and 6 respectively, but for *T. sacculifer* and using the equation of Mulitza et al.

(2003). Note that the basic patterns indicative of habitat tracking remain, but that the general calcification depth appears greater, also at lower temperatures.

Fig. 9: Effect of habitat tracking: reduced magnitude of deglacial temperature change estimated from Mg/Ca of *G. ruber* (pink) (Elderfield and Ganssen, 2000) compared to faunal assemblage based

seasonal temperature estimates (Chapman et al., 1996) in the subtropical North Atlantic. The predicted *G. ruber* (pink) temperature, which is similar to the Mg/Ca temperature, is based on the relationship identified in Figure 2 and the assemblage-derived temperatures. Values are anomalies with respect to the 0-10,000 years BP average.

Fig. 10: Conceptual model of calcification habitat change for warm and cold-water species. The coloured plane indicates the average calcification season and depth as a function of latitude. Dashed lines on top highlight the change in the seasonality.

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



Fig. 1

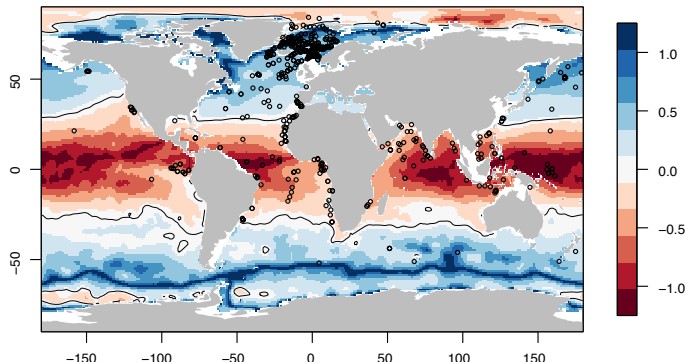





Fig. 2

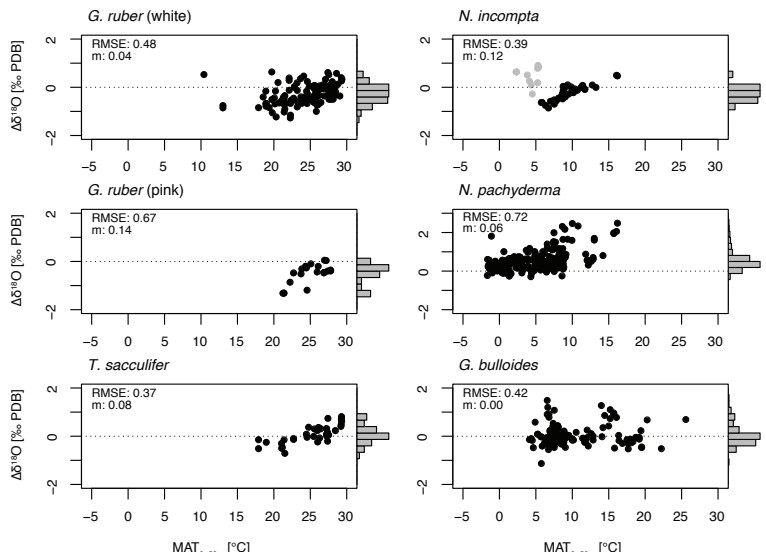





Fig. 3

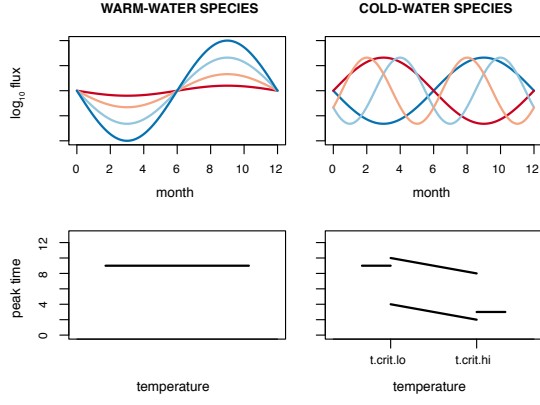





Fig. 4

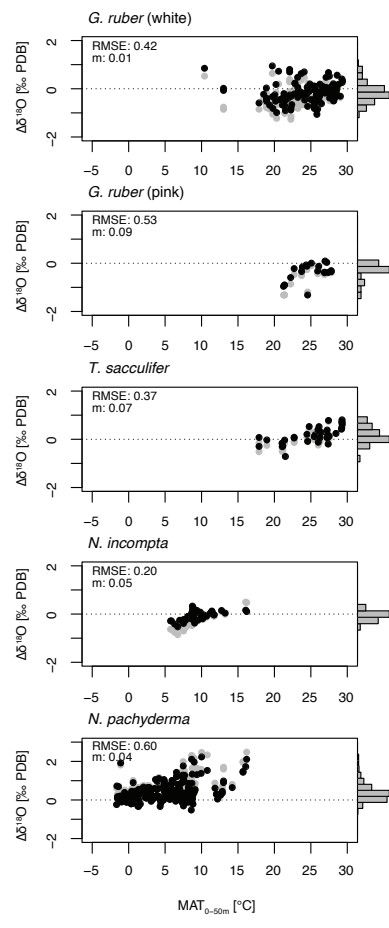





Fig. 5

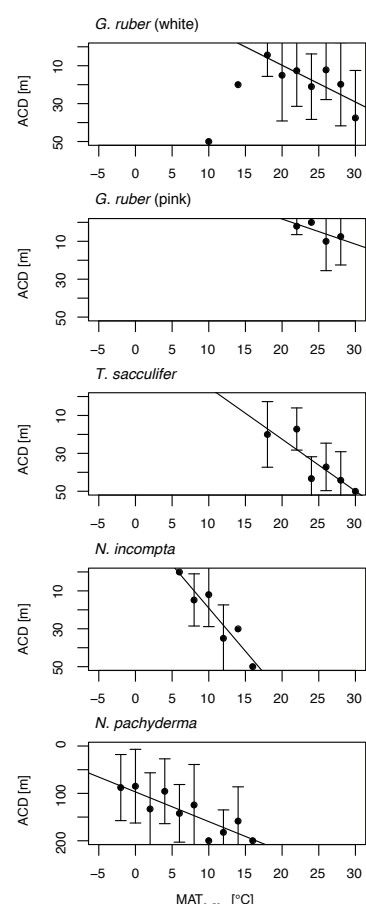






Fig. 6

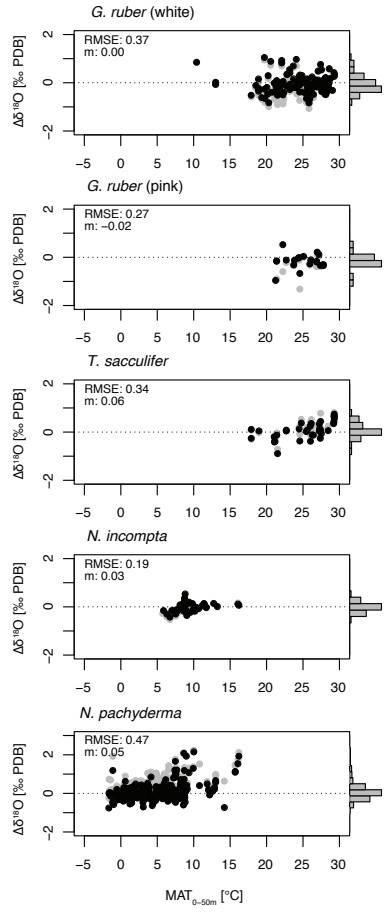





Fig. 7

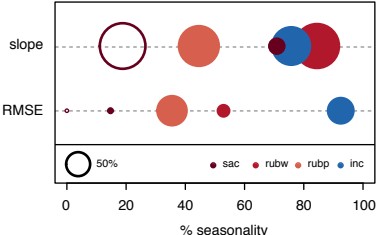


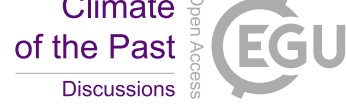

Fig. 8

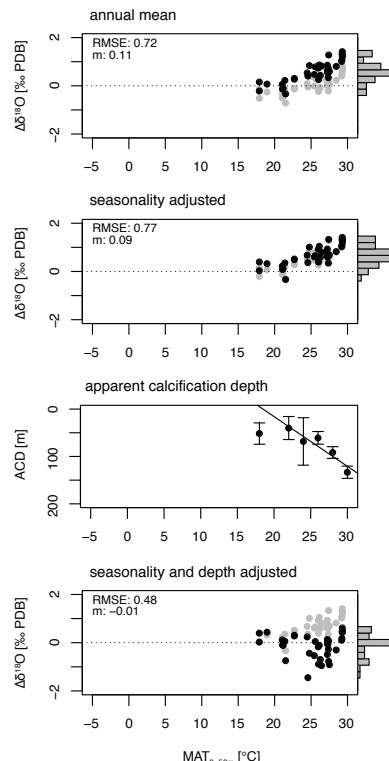





Fig. 9

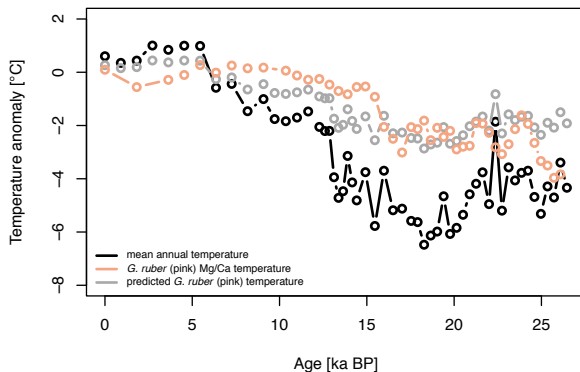



Fig. 10

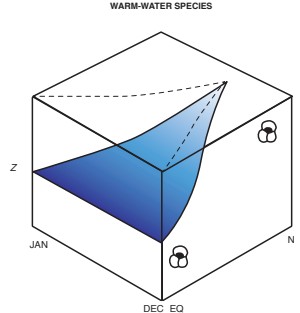
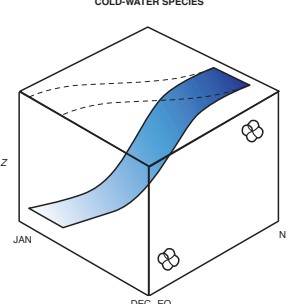
