# Peer review of "Quantifying the effect of seasonal and vertical habitat tracking on planktonic foraminifera proxies"

_Climate of the Past, 2016_

## Referee Comment (RC1) · Anonymous Referee #1 · 8 Feb 2017

The manuscript of Jonkers and Kucera on "Quantifying the effect of seasonal and vertical habitat tracking on planktonic foraminifera proxies" reads good. However, the topic and the findings discussed in the manuscript are not new. Some of the newest findings on the seasonal production of species were possibly published earlier by another member of the same working group (Kretzschmer et al. 2016). Having said this, I still like the presentation of the results and new figures providing an interesting perspective of a familiar problem. I would guess that the manuscript can be published with minor revisions. Please find my comments below.

Line 68, seasonality, please refer Kretzschmer et al. (2016) Line 140, G. bulloides, does include all genotypes and morphotypes in this case? Please clarify. In line 296,

genotypes are mentioned and may be discussed here. Line 160, Southern, in lower case. Lines 224-229: Using a sine function is not imperative. Could this not be improved in the present manuscript, in comparison to Jonkers and Kucera (2015)? Line 243, Discussion: to my impression, the discussion has started long before. Reorganize chapters? Line 249-250, "role" of temperature and "dominant in controlling": better replace "role" and "dominant in controlling" by "correlation" and "related to". The effect of temperature is neither proven nor quantified, since it rather effects foraminifera indirectly through oxygen concentration, and other variables. Lines 267, 310, 379: Referring as to "homeostatic . . . / homeostasy" in protists sounds wrong to me. Please delete in lines 267 and 379, and reword in line 310. Line 314: "constant seasonality and depth habitat", please give a reference. Lines 319-325: This is not new, and the authors try to convince the choir. Line 328-329: ". . . because temperature change forces the foraminifera to live in a seasonal or vertical 'window' ". What is the difference to Kretzschmer et al. (2016)? Caption Fig. 10: Please explain: depth (z) In general, figures could need some more explanation.

———————————————————

---

## Referee Comment (RC2) · Anonymous Referee #2 · 15 Feb 2017

Review for Climate of the Past of: "Quantifying the effect of seasonal and vertical habitat tracking on planktonic foraminifera proxies". By L. Jonkers and M. Kuçera.

The oxygen isotope composition in planktonic foraminifera is considered to be primarily a function of the ambient temperature and the oxygen isotope composition of seawater ($\delta18Ow$) during calcification. To a lesser extent the oxygen isotope composition of the tests ($\delta18Oforam$) may change as a function of other environmental factors related to the (carbonate) chemistry of the seawater or biological controls. Previous studies have indicated that the production of planktonic foraminifera is not distributed uniform over the time span of a year, but that growth takes place during a season in which most suitable conditions prevail. The same holds for the calcification depth, which may vary

depending on the water column conditions such as temperature, (temperature or physical) structure (e.g. stratification) or may depend on the depth of food availability. As such planktonic foraminifera may have a changing season of growth and depth, both depending on the local / regional ocean-climate conditions. The study presented by Jonkers and Kuçera deals with the goal to unravel the seasonal/depth signal recorded in the oxygen isotope composition of planktonic foraminifera. The study aims quantification of seasonal & depth habitat tracking by some species of planktonic foraminifera. Oxygen isotope measurements from core top sediments, are used in this study to obtain insight in the habitat tracking of foraminifera.

The manuscript generally reads well, is sufficiently illustrated and referenced. Unfortunately, the data used / presented seriously lack estimates of variability and appropriate statistical testing. At this stage it is unclear if the same conclusion will be reached after more careful consideration of the error and variability of the data used. It is recommended to include an analysis of variability and error and investigate the consequences of this for the significance of the regressed slopes used on which the main conclusion is based. As such the manuscript in its present form is suitable for publication after revisions.

Comments and suggestions

In general readability / clarity can be improved by associating $\delta$18O or $\Delta\delta$18O with the appropriate subscript i.e. indicating water(w), foram (f) or equilibrium(eq), foram minus water (f-w) or foram minus equilibrium ($\Delta\delta$18O$_{foram-eq}$) etc. At times this is unclear or even lacking, and therefore confusing and obstructing smooth reading.

In the first part of the study, the authors have investigated whether there is a trend between the $\delta$18O$_{foram}$ - $\delta$18O$_{eq.am.0-50m}$ - briefly referred to as $\Delta\delta$ 18O - and the mean annual temperature (MAT). If there is a trend, it can be concluded that species have recorded temperatures systematically deviating from the MAT. The analysis shown is key to the conclusion of the study, and it because of this importance and

further implications for the study that more transparency is needed in showing these data, in combination with a more appropriate and sound statistical assessment.

Looking at the data represented in Figure 2, it is not clear what the original ($\delta18$Oforam and $\delta18$Oequilibrium) data are alike. The reason for this is that - in Figure 2 - only the difference between the foraminiferal $\delta18$Of and the equilibrium value is graphically provided. I would say this showing the ("raw") data used for a study like this would be the first thing to do. Why I think this is important? It is so because it can be expected that the equilibrium value changes as a function of upper ocean temperature and $\delta18$Ow and error associated with the data can be made visible (see below). As such, it is recommended to insert a new figure - between the present Figure 1 and 2 - showing the foraminiferal $\delta18$O measurements and their associated equilibrium values including estimates of error / variability in the form of error bars (e.g. s.d. or c.i. intervals)!!

The data discussed and graphically represented in Figure 2 presently lack estimates of variability, that is, the data shown are not associated with an estimate of variability resulting from (measurement) error, environmental (i.e. seasonal) temperature variability (MAT is used, but the degree to which MAT is known varies as a function of seasonality), and variability resulting from the $\delta18$Ow estimates (expected to be relatively high at high latitudes and used to calculate the $\delta18$Oeq.am.0-50m) for which regressions vs. salinity - with error - have been used (LeGrande and Schmidt (2006). It is only later in the manuscript the authors refer to this as "some inherent noise" (line 206) indicating that the authors are aware their data should be associated with s.d. or alternatively with a confidence intervals to properly assess the information. The point here I want to make is that I disagree with the statement that this would be "...some inherent noise...". There is unfortunately, no effort or attempt made to provide any form of error / variability assessment, while in my view there's plenty of opportunity to do so (SST variability is known since atlas data were used, error in $\delta18$Ow can be assessed via the regressions used etc. etc.). The assessment of the variability is key to the conclusion that

(line 143): "...five out six analyzed species appear to minimise experienced temperature / environmental change, consistent with our hypothesis that....". Just reporting the RMSE and intercept of the regression is not sufficient to support the hypothesis that several species show evidence of habitat tracking. The authors should make a serious effort to come up with a decent quantification of error and convincingly show that the conclusion drawn from the data is statistically sound and robust! Once "x y" variability is assessed, an appropriate statistical test can be used to find out whether the slopes shown in Figure 2 are indeed significantly different from 'zero.

In section 4 "seasonality", the log (flux) pattern is described as "...a sine wave of which the amplitude and phasing are changed as a function of the annual mean temperature...". Although this may - intuitively - be a reasonable approximation for the extra-tropics, I wonder if the approach followed also agrees with the flux patterns for species living in the tropical oceans where insolation is not a limiting factor and there is two maxima in the solar insolation during the course of a year. It seems that the authors do observe a problem with this model in predicting the seasonal flux pattern of species in the tropics (lines 154- 157), but it is not explained / clarified why this is so and what the implication would be for their conclusion! Likely the seasonality of species in the tropics is driven by other factors than temperature? Maybe this can be clarified better in the context of Lombard et al., (2009) (Mar.Mic, 70, 1-7) and Lombard et al., (2011) (Biogeosciences, 8, 853-873), where species growth rates are modelled as a function of temperature. If using an ocean model in combination with temperature dependent growth rates i.e. using an ecophysiological model, one can likely predict the oxygen isotope composition reasonably well. I wonder why such an approach, i.e. using an eco-physiological model, has not been chosen and preference is given to modelling the flux as "a function of a shifted the sine wave"? This should be clarified.

Similarly to the remarks above for Figure 2 and associated data, data shown in Figure 5 data should have variability indication. A test of slope should be performed to show the slope is not significantly different from "zero', further supporting the hypothesis that

depth habitat migration may indeed occur.

line 63: Should read: "...a clear relationship with sea surface temperature .." line 64: Sentence unclear, consider rephrasing: "While the latter trend....will reflect". line 75: Vertical habitat? Recommended to change to "depth habitat". line 79: Geochemical data: mention Mg/Ca, $\delta$18Oforam. Line 81: Start new paragraph. line 90: Seasonal sea surface temperature instead of seasonal temperature. line 97: dampening effect: i.e. reduction of the recorded range versus the environmental (observed) range. line 100: "foraminifera proxies". Better: "foraminiferal $\delta$18O" line 133: Incomplete. Change "..high temperatures..." into "..high annual mean temperatures..". line 133: Change "...higher calcification temperatures..." into "...higher than annual mean calcification temperatures...". line 137: "...Nordic Seas outside of the direct...". Remove "of". line 136-139: "These observations...further analysis". This sentence is quite long. Consider making two. Second sentence may start after North Atlantic Drift. line 140-141: "..is the only species that..." can be removed. line 143: "...analysed species.." may be changed into "...species analysed...". line 163: "all of" can be removed

line 202: "Our analysis allows partitioning of habitat change in to changes in seasonality and calcification depth for .....". If statistically robust, and the same conclusion holds after analysis of variability, the analysis still does not allow (a full) partitioning in my opinion. I recommend to phrase more careful. line 204: use $\delta$18Oforam instead of just $\delta$18O. Note that the delta notation has been used in two forms. Indicate which one is applicable.

line 244-252: The effect of an nutrient depleted mixed layer quite typical for the tropical ocean structure is not considered as an option for deeper & colder growth. Simply the fact that species can find their food deeper in the water column (Deep Chlorophyl Maximum), just because the mixed layer is nutrient poor and as such contains less particulate matter, is not considered here. It would offer a very plausible explanation for deeper growth - at lower than SST - in the tropics.

line 261: "...foraminifera grow their test exponentially..." needs rephrasing. i.e. " ...shell mass increases exponentially as a function of shell size... "

line 303: The remark that the species $\Delta\delta$18O "at face value" holds the best promise of providing reconstructions of mean annual near surface conditions is may be a bit mystifying. As is explained in the next section, G. bulloides is characteristic for high nutrient waters and in (tropical) upwelling systems the species is associated with upwelling and hence calcifying at or close to the lowest SST's during the year. Since the SST's during upwelling are deviating from AM conditions, it may be better to say that right away that G. bulloides does not reflect AM conditions.

line 314: "assumption of constant seasonality and depth habitat" references? or leave out...

line 322: Rephrasing needed: "..not driven by mean annual temperature..". How can a mean temperature drive anything?? The mean is a statistic!

line 379: homeostatic behaviour? I'm quite sure that the term 'homeostasis' is applicable to humans/warm blooded animals. I guess there is not such a regulatory system present in uni-cellular zoo-planktic algae! I guess the ability of foraminifera to potentially "actively" seek optimal conditions may more have to do with their genetic and epigenetic (not investigated so far) profiles.

Please provide more informative Figure captions these are very brief!

———————————————————

---

## Editor Comment (EC1) · L. Skinner (Editor) · 5 Apr 2017

Dear Lukas and Michal,

I would like to invite you to prepare a full response to the review comments received so far, accompanied by a suitably revised manuscript if you so wish. These review comments cover a number of specific points, which you should strive to address. However they also dwell on two main themes: the need for more thorough statistical analyses; and the need to place your study more squarely in the context of previous work. I would add to this a further proposal to delve deeper into a discussion of what ultimately can be achieved in light of your results, and to add clear detail to your proposals regarding the possibility (if indeed this exists) of improving the accuracy of palaeoceanographic

reconstructions and/or data-model comparisons. My impression is that the latter are currently only vaguely described.

The issue of relevant previous work is always a tricky issue to address, given the depth and interpretative nature of the literature; however I do tend to concur with the comment that the manuscript seems often to 'preach to the converted' regarding the importance of taking into account habitat bias in the interpretation of foraminifer proxy records, in particular for planktonic species. I also agree that for a paper on such a well-known (if also often completely ignored) issue, the literature review is rather sparse. I would therefore ask you to consider taking this opportunity in revising your manuscript to amplify on this aspect, e.g. for the benefit of readers who will be well aware of the issue, but perhaps not all of the key literature on the topic.

Of course, a full literature review of a topic is not a prerequisite for any given paper on that topic (and is not what I would propose you undertake); however, I suspect that in this case the issue comes to the fore in light of the apparent lack of progress in developing a 'solution' for dealing with the issue of habitat bias in planktonic foraminifera. In this regard, one might argue that there are two schools of thought: one that proposes to 'fix' foraminifer proxy records by correcting for habitat biases; and another that proposes to accept them (along with their ultimately cryptic nature) and to detect and/or 'employ' them as they arise, even if they cannot be known a priori. It seems that the manuscript currently makes a strong point that habitat biases are real (which arguably we already knew), as well as much weaker point regarding what is to be done (or can be done) in light of their existence. I would encourage you to consider engaging in a deeper discussion of the latter issue, as I think this would lend the manuscript greater force.

In order to illustrate my proposal, I would refer to the exclusion of G.bulloides from the analysis: this species does not show the 'fingerprint' of habitat bias as defined in the manuscript, and yet it does have a clear habitat bias. This may illustrate a shortcoming of the approach taken, with respect to accounting for habitat bias (or how

Interactive
comment

it has been 'modeled'); namely that it does not address habitat bias that results in fixed deviations from the annual mean for example. Further concerns might arise with respect to accounting for habitat bias when considering proxy reconstructions of past climatic change and the occurrence of 'non-analogue' situations (e.g. strong seasonal perturbations, or stratification changes). Can your analysis be expanded to consider in more detail 'what can be done', beyond recognizing that habitat bias is an issue? Would *parallel* multi-proxy and multi-species analyses be helpful for example (e.g. as a set of 'parallel equations' for habitat variability within a relatively constrained yet still unknown habitat range)?

I hope that you will find all of these comments helpful, and I look forward to receiving your revised manuscript and response.

Sincerely, Luke Skinner

---

## Author Comment (AC1) · 20 Apr 2017

Dear Luke,

We would like to thank to the reviewer for her/his comments that helped to improve the manuscript and sharpen our argument that habitat tracking deserves more attention from the paleoceanographic community. Below we respond to the comments in red. Line numbers refer to the version with tracked changes that we have appended to our response to your comment.

We hope that our revised manuscript now meets the criteria for publication in Climate of the Past.

Lukas Jonkers and Michal Kucera
* * *
The manuscript of Jonkers and Kucera on "Quantifying the effect of seasonal and vertical habitat tracking on planktonic foraminifera proxies" reads good. However, the topic and the findings discussed in the manuscript are not new. Some of the newest findings on the seasonal production of species were possibly published earlier by another member of the same working group (Kretzschmer et al. 2016). Having said this, I still like the presentation of the results and new figures providing an interesting perspective of a familiar problem. I would guess that the manuscript can be published with minor revisions. Please find my comments below.

Before we respond in detail to the comments, we would like to shortly address the novelty issue of our manuscript. We realise that the phenomenon of habitat tracking a appears familiar and unsurprising. Its existence has been invoked (or can be anticipated) from previous work on plankton tow and sediment trap samples. However, studies explicitly demonstrating the kind and size of the habitat bias in foraminifera proxies are rare and attempts to quantify the effect are – to the best of our knowledge – virtually non-existent. This observation is supported by the nature of the comments pertaining to the novelty of our study: the referees perceive the results as unsurprising and perhaps not novel, but do not provide references to previous work that used empirical data to investigate habitat tracking. We agree that the issue of habitat tracking is logical (and to be expected), but to the best of our knowledge it has not been given an explicit and quantitative treatment before. Indeed, this is the point where we believe the novelty of our study lies: not in the suggestion that habitat tracking affects foraminifera proxy records, but demonstrating that this occurs in a predictable manner and attempting to quantify the effect.

Line 68, seasonality, please refer Kretzschmer et al. (2016). Will do.
Line 140, G. bulloides, does include all genotypes and morphotypes in this case? Please clarify. In line 296, genotypes are mentioned and may be discussed here.
This is a valid point that applies not only to this species. All of the stable isotope data we analyse have been generated without considering the existence of multiple genetic types. We will add this information to the 'Data and approach' section of the manuscript. The section around line 140 of the original submission is a description of the first results and the information would not be best placed here. We also prefer to keep the

discussion about why *G. bulloides* may show a different pattern from the other species separate from the results section.

Line 160, Southern, in lower case. This is Copernicus policy.

Lines 224-229: Using a sine function is not imperative. Could this not be improved in the present manuscript, in comparison to Jonkers and Kucera (2015)?

Approximating flux seasonality by a sine wave is a simple and mathematically elegant solution, but we fully agree with the referee that it is not the only possible model of the reality (which may have a different underlying dynamics). We chose to use this notation, because in previous work we have demonstrated that seasonal flux patterns of foraminifera can be well described by a sine function (Jonkers and Kučera, 2015). Since it is not a priori clear that other approaches would yield better results, development of a new seasonality (or habitat) model goes beyond the scope of this study. We do take this suggestion as encouragement to continue ongoing research into the prediction of foraminifera habitat.

Line 243, Discussion: to my impression, the discussion has started long before. Reorganize chapters?

We agree with the reviewer that the sections preceding the discussion section do also contain some discussion and have therefore renamed the section headers.

Line 249-250, "role" of temperature and "dominant in controlling": better replace "role" and "dominant in controlling" by "correlation" and "related to". The effect of temperature is neither proven nor quantified, since it rather effects foraminifera indirectly through oxygen concentration, and other variables.

We agree with the suggestion to reword.

Lines 267, 310, 379: Referring as to "homeostatic . . . / homeostasy" in protists sounds wrong to me. Please delete in lines 267 and 379, and reword in line 310. Will do.

Line 314: "constant seasonality and depth habitat", please give a reference.

Rather than arbitrarily singling out a few studies among the many that ignore habitat variability, we prefer to highlight one of the few studies that acknowledges the issue (lines 457-458).

Lines 319-325: This is not new, and the authors try to convince the choir.

Indeed, the assertion of this sentence is not new. It was not our intention to make is sound as if it were. As we note in the first comment, we realise that the existence of this issue is likely considered as well known and accepted by the community. It is here used as an opening statement for a section where we discuss the consequences of this phenomenon in view of the quantitative models we developed.

Line 328-329: ". . . because temperature change forces the foraminifera to live in a seasonal or vertical 'window' ". What is the difference to Kretzschmer et al. (2016)?

The difference is in the approach. Kretschmer et al. (2016) and indeed the preceding studies by Fraile et al. (2008) and Lombard et al. (2011) are all based on numerical models, where the seasonal habitat of a species is the result of the parametrisation of its ecological preferences. If temperature preference is set to warm that the species in the model will grow only where and when "warm" is realised. This approach is powerful in testing the likely behaviour of the species under conditions that do not occur today. However, this approach provides no constrains on whether the species really behave in this way. In contrast, in this study, we provide empirical evidence that habitat tracking affects proxies and derived empirical estimates of the size of this effect.

Caption Fig. 10: Please explain: depth (z). In general, figures could need some more explanation.

We will replace z with depth to avoid confusion and expand on the figure captions.

Fraile, I., Schulz, M., Mulitza, S., and Kucera, M.: Predicting the global distribution of planktonic foraminifera using a dynamic ecosystem model, Biogeosciences, 5, 891-911, 10.5194/bg-5-891-2008, 2008.
Jonkers, L., and Kučera, M.: Global analysis of seasonality in the shell flux of extant planktonic Foraminifera, Biogeosciences, 12, 2207-2226, 10.5194/bg-12-2207-2015, 2015.
Lombard, F., Labeyrie, L., Michel, E., Bopp, L., Cortijo, E., Retailleau, S., Howa, H., and Jorissen, F.: Modelling planktic foraminifer growth and distribution using an ecophysiological multi-species approach, Biogeosciences, 8, 853-873, 10.5194/bg-8-853-2011, 2011.

---

## Author Comment (AC2) · 20 Apr 2017

Dear editor, dear Luke,

Please find below our response to the comments by reviewer 2. We are grateful for the suggestion to look deeper into the uncertainty associated with the data. We have carefully taken the issue into account and believe that the updated analysis strengthens the main message of our study.

Below we respond to the comments in red. Line numbers refer to the version of the manuscript with highlighted changes (appended to our response your comment). We hope that this revised version merits publication in Climate of the Past.

Kind regards,

Lukas Jonkers and Michal Kucera
* * *
Review for Climate of the Past of: "Quantifying the effect of seasonal and vertical habitat tracking on planktonic foraminifera proxies". By L. Jonkers and M. Kuçera.

The oxygen isotope composition in planktonic foraminifera is considered to be primarily a function of the ambient temperature and the oxygen isotope composition of seawater (18Ow) during calcification. To a lesser extent the oxygen isotope composition of the tests (18Oforam) may change as a function of other environmental factors related to the (carbonate) chemistry of the seawater or biological controls. Previous studies have indicated that the production of planktonic foraminifera is not distributed uniform over the time span of a year, but that growth takes place during a season in which most suitable conditions prevail. The same holds for the calcification depth, which may vary depending on the water column conditions such as temperature, (temperature or physical) structure (e.g. stratification) or may depend on the depth of food availability. As such planktonic foraminifera may have a changing season of growth and depth, both depending on the local / regional ocean-climate conditions. The study presented by Jonkers and Kuçera deals with the goal to unravel the seasonal/depth signal recorded in the oxygen isotope composition of planktonic foraminifera. The study aims quantification of seasonal & depth habitat tracking by some species of planktonic foraminifera. Oxygen isotope measurements from core top sediments, are used in this study to obtain insight in the habitat tracking of foraminifera.

The manuscript generally reads well, is sufficiently illustrated and referenced. Unfortunately, the data used / presented seriously lack estimates of variability and appropriate statistical testing. At this stage it is unclear if the same conclusion will be reached after more careful consideration of the error and variability of the data used. It is recommended to include an analysis of variability and error and investigate the

consequences of this for the significance of the regressed slopes used on which the main conclusion is based. As such the manuscript in its present form is suitable for publication after revisions.

Comments and suggestions

In general readability / clarity can be improved by associating 18O or 18O with the appropriate subscript i.e. indicating water(w), foram (f) or equilibrium(eq), foram minus water (f-w) or foram minus equilibrium (18Oforam-eq) etc. At times this is unclear or even lacking, and therefore confusing and obstructing smooth reading.

We will make sure to add appropriate subscripts to enhance clarity.

In the first part of the study, the authors have investigated whether there is a trend between the 18Oforam - 18Oeq.am.0-50m - briefly referred to as 18O – and the mean annual temperature (MAT). If there is a trend, it can be concluded that species have recorded temperatures systematically deviating from the MAT. The analysis shown is key to the conclusion of the study, and it because of this importance and further implications for the study that more transparency is needed in showing these data, in combination with a more appropriate and sound statistical assessment.

This is a perfectly accurate description of the starting point of our study. This seemed so obvious to us that we have omitted this step in the figures and started by analysing the residual structure. We understand the merit of providing a more basal evaluation of the original data (including uncertainties, see below) and propose to include these in the supplement.

Looking at the data represented in Figure 2, it is not clear what the original (18Oforam and 18Oequilibrium) data are alike. The reason for this is that - in Figure 2 – only the difference between the foraminiferal 18Of and the equilibrium value is graphically provided. I would say this showing the ("raw") data used for a study like this would be the first thing to do. Why I think this is important? It is so because it can be expected that the equilibrium value changes as a function of upper ocean temperature and 18Ow and error associated with the data can be made visible (see below). As such, it is recommended to insert a new figure - between the present Figure 1 and 2 - showing the foraminiferal 18O measurements and their associated equilibrium values including estimates of error / variability in the form of error bars (e.g. s.d. or c.i. intervals)!!

Next to providing the figure of the raw data, we have followed the suggestion of the referee and attempted to estimate the uncertainties in the various d18O values. To this end, we consider the following main sources of uncertainty:

- Uncertainty on the observed d18Ocalcite values based on the standard deviation of repeat measurements in the MARGO dataset. This amounts to 0.12 ‰.
- A calibration uncertainty with respect to predicted d18O. We use 0.2 ‰ based on previous work.

- Uncertainty associated with the predicted d18Oeq, which in our opinion is mainly driven by uncertainty in the estimation of d18Osw from salinity. This uncertainty varies regionally and is largest in the Arctic, where it reaches 0.91 ‰.

We propagate these uncertainties using a Monte Carlo approach; details are described in section 2: Data and approach (lines 177-182). Uncertainty on the mean annual temperature and salinity values is not taken into account because these are based on many observations, rendering the error negligible. Since at this place of the argument our zero hypothesis is that foraminifera record annual mean conditions (not any month or season within a year), we can ignore the uncertainty associated with intra-annual temperature and salinity variability.

The resulting uncertainty estimates support our original conclusion that five out of six of the species do not record mean annual conditions in the upper water column (Sup Fig. 1). Since the new figure and our original figure 2 are partly redundant, we chose to include the new figure in the supplementary information.

The data discussed and graphically represented in Figure 2 presently lack estimates of variability, that is, the data shown are not associated with an estimate of variability resulting from (measurement) error, environmental (i.e. seasonal) temperature variability (MAT is used, but the degree to which MAT is known varies as a function of seasonality), and variability resulting from the 18Ow estimates (expected to be relatively high at high latitudes and used to calculate the 18Oeq.am.0-50m) for which regressions vs. salinity - with error - have been used (LeGrande and Schmidt (2006). It is only later in the manuscript the authors refer to this as "some inherent noise" (line 206) indicating that the authors are aware their data should be associated with s.d. or alternatively with a confidence intervals to properly assess the information. The point here I want to make is that I disagree with the statement that this would be "...some inherent noise...". There is unfortunately, no effort or attempt made to provide any form of error / variability assessment, while in my view there's plenty of opportunity to do so (SST variability is known since atlas data were used, error in 18Ow can be assessed via the regressions used etc. etc.). The assessment of the variability is key to the conclusion that (line 143): "...five out six analyzed species appear to minimise experienced temperature / environmental change, consistent with our hypothesis that....". Just reporting the RMSE and intercept of the regression is not sufficient to support the hypothesis that several species show evidence of habitat tracking. The authors should make a serious effort to come up with a decent quantification of error and convincingly show that the conclusion drawn from the data is statistically sound and robust! Once "x y" variability is assessed, an appropriate statistical test can be used to find out whether the slopes shown in Figure 2 are indeed significantly different from 'zero.

The referee is right to demand such analysis. Please also see our response above. We have followed this approach to estimate the error of the Dd18O-MAT relationships (Fig. 2). The error envelopes show the 5 to 95 percentiles of the Monte Carlo analysis and confirm our original conclusion that there is a

relationship between MAT and the offset from annual mean d18O in the upper water column in 5 out of 6 species, which is consistent with the expected effect of habitat tracking.

In section 4 "seasonality", the log (flux) pattern is described as "...a sine wave of which the amplitude and phasing are changed as a function of the annual mean temperature...". Although this may - intuitively - be a reasonable approximation for the extratropics, I wonder if the approach followed also agrees with the flux patterns for species living in the tropical oceans where insolation is not a limiting factor and there is two maxima in the solar insolation during the course of a year. It seems that the authors do observe a problem with this model in predicting the seasonal flux pattern of species in the tropics (lines 154- 157), but it is not explained / clarified why this is so and what the implication would be for their conclusion! Likely the seasonality of species in the tropics is driven by other factors than temperature? Maybe this can be clarified better in the context of Lombard et al., (2009) (Mar.Mic, 70, 1-7) and Lombard et al., (2011) (Biogeosciences, 8, 853-873), where species growth rates are modelled as a function of temperature. If using an ocean model in combination with temperature dependent growth rates i.e. using an ecophysiological model, one can likely predict the oxygen isotope composition reasonably well. I wonder why such an approach, i.e. using an eco-physiological model, has not been chosen and preference is given to modelling the flux as "a function of a shifted the sine wave"? This should be clarified.

The primary goal of our submission was to show that habitat tracking influences fossil signal and by how much – to the best of our knowledge this has not been demonstrated in this way before - and to raise awareness in the paleoceanographic community that the issues should be taken into account. Once showing that the sedimentary isotopic signal bears a signature of habitat tracking, we face the inherently more difficult question of the exact attribution of the habitat tracking to depth and season. In this study, we addressed the problem by constraining seasonality through observations. This is in our opinion the best approach because there are much better data on seasonality than on depth habitat. To this end, we adopted a sine wave model of flux seasonality based on previous work where we showed that foraminifera flux patterns can be described using a simple sine wave and that modulation of this sine wave (amplitude and phasing) can be predicted by temperature (Jonkers and Kučera, 2015). While we agree with the reviewer that other (temperature-related) factors are likely to be important too and that different approaches exist to model foraminifera seasonality (Lombard et al., 2011; Fraile et al., 2008), we decided to stick to a formulation that is entirely based on empirical observations. This is acknowledged in the manuscript on lines 237-238 Please also refer to our response to a similar comment by reviewer 1 for a motivation of our model choice.

As mentioned in the original text (lines 229-231), the inability of our model to capture the random flux peak timing in the tropics is of negligible consequence for the sedimentary signal because in the tropics both shell fluxes (low amplitude, or peak prominence as described in Jonkers and Kucera (2015)) and SST are relatively constant during the year.

Similarly to the remarks above for Figure 2 and associated data, data shown in Figure 5 data should have variability indication. A test of slope should be performed to show the slope is not significantly different from "zero', further supporting the hypothesis that depth habitat migration may indeed occur.

We understand this point, but note that this can only be addressed indirectly. This is because in the moment we included the modelled seasonal effect on the d18O, we have added a source of uncertainty that is hard to constrain. The residual Dd18O after seasonal weighting of the d18Oeq is model dependent and the model has parameters with unconstrained uncertainty (in fact the formulation of the model itself – using a sine wave is not certain). This is why we can only proceed with the analyses as shown in figure 5, "given" the particular model formulation. This means that effectively we ask the question 'is the residual Dd18O after seasonality correction depth dependent when we use this particular seasonality model?'.

However, we feel we owe the readers at least a first order estimate of the sensitivity of the result on the parameters of the sine-wave model we use. . We have therefore explored how sensitive the apparent calcification depth (ACD)-temperature relationship is to the slope and intercept of the MAT-flux amplitude relationship.

We show the results for G. ruber pink (Sup fig.2), obtained by doubling and halving the slope and intercept of the MAT-flux relationship with respect to the empirical values obtained from (Jonkers and Kučera, 2015). Increasing the seasonality reduces the RMSE and the dependency of Dd18O on MAT in the seasonally weighted Dd18O estimates. It yields estimates of ACD that appear not or positively correlated with MAT and leads to (seasonality and depth weighted Dd18O) RMSE and Dd18O-MAT slopes similar to the observation-based model.

The reverse holds true for a reduction in seasonality, which yields RMSE larger and Dd18O-MAT slopes steeper than when using mean annual values and implies Dd18O-MAT relationships similar to our original seasonality-only Dd18O estimates and RMSE close to the Dd18O based on annual mean values.

This suggests that the formulation of seasonality in our model is conservative: weaker seasonality parametrisation leaves much larger residuals and a slope that cannot be accounted for by depth habitat adjustment. However, we note that in the case of G. ruber pink there exists a parametrisation of flux seasonality that leads to a greater improvement in the d18O prediction and implies a constant habitat depth adjustment.

We will add the discussion above to the section 'seasonality vs. depth habitat'.

The following minor comments have all been addressed/changed. We have provided a response only in cases where we don't agree with the reviewer or feel that more explanation is needed.

line 63: Should read: "...a clear relationship with sea surface temperature .."

line 64: Sentence unclear, consider rephrasing: "While the latter trend....will reflect".

line 75: Vertical habitat? Recommended to change to "depth habitat".

line 79: Geochemical data: mention Mg/Ca, 18Oforam.

Line 81: Start new paragraph

line 90: Seasonal sea surface temperature instead of seasonal temperature. Not strictly sea surface, so we prefer not to change the wording here.

line 97: dampening effect: i.e. reduction of the recorded range versus the environmental (observed) range.

Line 100: "foraminifera proxies". Better: "foraminiferal 18O" We prefer to keep the original general wording since to our knowledge no other study has looked into this effect.

line 133: Incomplete. Change "..high temperatures..." into "..high annual mean temperatures..".

line 133: Change "...higher calcification temperatures..." into "...higher than annual mean calcification temperatures...".

line 137: "...Nordic Seas outside of the direct...". Remove "of".

Line 136-139: "These observations...further analysis". This sentence is quite long. Consider making two. Second sentence may start after North Atlantic Drift.

line 140-141: "..is the only species that..." can be removed. We would like to point out that G. bulloides is in our analysis the exception and hence prefer to keep the original.

line 143: "...analysed species.." may be changed into "...species analysed...".

line 163: "all of" can be removed

line 202: "Our analysis allows partitioning of habitat change in to changes in seasonality and calcification depth for .....". If statistically robust, and the same conclusion holds after analysis of variability, the analysis still does not allow (a full) partitioning in my opinion. I recommend to phrase more careful.

line 204: use 18Oforam instead of just 18O. Note that the delta notation has been used in two forms. Indicate which one is applicable.

line 244-252: The effect of an nutrient depleted mixed layer quite typical for the tropical ocean structure is not considered as an option for deeper & colder growth. Simply the fact that species can find their food deeper in the water column (Deep Chlorophyl Maximum), just because the mixed layer is nutrient poor and as such contains less particulate matter, is not considered here. It would offer a very plausible explanation for deeper growth - at lower than SST - in the tropics.

line 261: "...foraminifera grow their test exponentially..." needs rephrasing. i.e. " ...shell mass increases exponentially as a function of shell size... "

line 303: The remark that the species 18O "at face value" holds the best promise of providing reconstructions of mean annual near surface conditions is may be a bit mystifying. As is explained in the next section, G. bulloides is characteristic for high nutrient waters and in (tropical) upwelling systems the species is associated with upwelling and hence calcifying at or close to the lowest SST's during the year. Since the SST's during upwelling are deviating from AM conditions, it may be better to say that right away that G. bulloides does not reflect AM conditions.

line 314: "assumption of constant seasonality and depth habitat" references? or leave out...

As explained in our response to reviewer 1 we prefer to give a positive example instead..

line 322: Rephrasing needed: "..not driven by mean annual temperature..". How can a

mean temperature drive anything?? The mean is a statistic!

line 379: homeostatic behaviour? I'm quite sure that the term 'homeostasis' is applicable to

humans/warm blooded animals. I guess there is not such a regulatory system present in uni-cellular zooplanktic algae! I guess the ability of foraminifera to potentially "actively" seek optimal conditions may

more have to do with their genetic and epigenetic (not investigated so far) profiles.

Please provide more informative Figure captions these are very brief!

Fraile, I., Schulz, M., Mulitza, S., and Kucera, M.: Predicting the global distribution of planktonic foraminifera using a dynamic ecosystem model, Biogeosciences, 5, 891-911, 10.5194/bg-5-891-2008, 2008.
Jonkers, L., and Kučera, M.: Global analysis of seasonality in the shell flux of extant planktonic Foraminifera, Biogeosciences, 12, 2207-2226, 10.5194/bg-12-2207-2015, 2015.
Lombard, F., Labeyrie, L., Michel, E., Bopp, L., Cortijo, E., Retailleau, S., Howa, H., and Jorissen, F.: Modelling planktic foraminifer growth and distribution using an ecophysiological multi-species approach, Biogeosciences, 8, 853-873, 10.5194/bg-8-853-2011, 2011.

---

## Author Comment (AC3) · 20 Apr 2017

The comment was uploaded in the form of a supplement:
http://www.clim-past-discuss.net/cp-2016-125/cp-2016-125-AC3-supplement.pdf

---

## Author Response (AR1)

Dear Luke,

Thank you for accepting our manuscript for publication in Climate of the Past. Following your comments we have made some changes to text and restructured the final section 'Implications and outlook' slightly to improve the readability of the manuscript.

Kind regards,

Lukas Jonkers and Michal Kucera